# Routine sub-2.5 Å cryo-EM structure determination of GPCRs

Radostin Danev [1✉], Matthew Belousoff[2,3], Yi-Lynn Liang[2,4], Xin Zhang[2,3], Fabian Eisenstein [1], Denise Wootten[2,3] & Patrick M. Sexton [2,3]

Cryo-electron microscopy (cryo-EM) of small membrane proteins, such as G protein-coupled receptors (GPCRs), remains challenging. Pushing the performance boundaries of the technique requires quantitative knowledge about the contribution of multiple factors. Here, we present an in-depth analysis and optimization of the main experimental parameters in cryo-EM. We combined actual structural studies with methods development to quantify the effects of the Volta phase plate, zero-loss energy filtering, objective lens aperture, defocus magnitude, total exposure, and grid type. By using this information to carefully maximize the experimental performance, it is now possible to routinely determine GPCR structures at resolutions better than 2.5 Å. The improved fidelity of such maps enables the building of better atomic models and will be crucial for the future expansion of cryo-EM into the structure-based drug design domain. The optimization guidelines given here are not limited to GPCRs and can be applied directly to other small proteins.

[1] Graduate School of Medicine, University of Tokyo, Tokyo, Japan. [2] Drug Discovery Biology, Monash Institute of Pharmaceutical Sciences, Monash University, Parkville, VIC, Australia. [3] ARC Centre for Cryo-electron Microscopy of Membrane Proteins, Monash Institute of Pharmaceutical Sciences, Monash University, Parkville, VIC, Australia. [4]Present address: Confo Therapeutics, Ghent (Zwijnaarde), Belgium. ✉email: rado@m.u-tokyo.ac.jp

The "revolution" in cryo-electron microscopy (cryo-EM) began a decade ago and led to substantial improvements in performance. It was initiated by the introduction of direct electron detectors and new data processing methods. Since then, there have been gradual technological advances, primarily on the data analysis side[1]. The majority of single particle reconstructions nowadays are in the 2.5–3.5 Å resolution range[2]. Such maps are adequate for unambiguous tracing of the backbone and positioning of most side chains, but depending on the case, the performance may be insufficient to confidently answer the research question. For example, understanding how small molecule ligands bind to and modulate receptors often requires the identification of not just direct interactions but also water molecule networks, and this is also critical for structure-based drug design. Resolutions better than 2.5 Å, and preferably better than 2.0 Å, where the nature of chemical interactions can be determined with precision, are necessary for structure-based approaches to be effective[3,4]. Ultimately, the level at which regions of a 3D cryo-EM map are resolved will be limited only by the thermal motions of the molecule. To get as close as practically possible to such performance, the experiment must be tuned for optimal signal extraction.

The outcome of a cryo-EM project depends strongly on the biochemical quality and behavior of the sample. Good particle homogeneity and concentration are important[1], but some targets may be difficult to purify in sufficient quantity or adopt a preferential orientation in thin ice. Obtaining a structure from such samples may require an experimental compromise that does not leave much room for optimization, such as data collection in thicker ice areas or with a tilted specimen[5]. The results and recommendations presented here are based on well-optimized samples and therefore may provide little or no benefit in such cases.

Until now, there have been only general guidelines about optimal cryo-EM data acquisition approaches. Each research group or cryo-EM facility has adopted a favored set of parameters based on their own experience, published results, and/or intuitive expectations. Some choices may be limited by the hardware configuration of the microscope, such as the accelerating voltage, the type of detector and the presence of an energy filter, while other parameters, such as support grid type, defocus range, total exposure and objective lens aperture (OLA), are decided by the researcher or the operator. Alone, each of these settings may have a small and seemingly insignificant contribution, but the combination of several optimal values has a cumulative effect and could lead to a noticeable improvement in map resolution and quality. Recent atomic resolution cryo-EM structures[6,7] demonstrated clearly that fine-tuning of the experiment is crucial for pushing the performance limits.

The first high-resolution cryo-EM structure of a G protein-coupled receptor (GPCR) complex was determined 4 years ago using the Volta phase plate (VPP)[8]. Active GPCR-transducer complexes are relatively small asymmetric assemblies (~150 kDa), which at the time were considered very challenging for cryo-EM. Data acquisition and image processing methods were not as advanced as they are today, and the capabilities of the conventional defocus phase contrast approach were not fully explored. Hoping to increase the probability of success, we decided to employ the VPP, and it produced remarkable results on the first try. Thereafter, we continued using the VPP in GPCR projects with great success[9–11], yielding results of similar or better quality than conventional defocus-based approaches[12–20]. Meanwhile, with increasing data acquisition throughput and new image processing methods, the quality of results from the conventional defocus approach continued to improve, even for molecules that are smaller than GPCRs[21]. Therefore, we decided to experimentally quantify the effect of the VPP for the study of GPCRs. To minimize the influence of other factors, datasets with and without the phase plate were collected on the same grid in a single cryo-EM session. All other experimental parameters were kept identical and therefore the VPP/non-VPP datasets could be joined to get a reconstruction for the structural study of the complex[22]. Single experimental parameter modification during data acquisition proved to be a very efficient way for conducting structural and methodological studies in parallel. It did not require additional microscope time and the comparative results were from a real-world sample. We used this tandem experimental approach to also evaluate the effect of zero-loss energy filtering and the OLA. Furthermore, we quantified the effect of defocus amount and total electron exposure by splitting one of the datasets into corresponding subsets.

Here, we present the results from the quantitative evaluation of the effect of the VPP, zero-loss energy filtering, OLA, defocus amount and total exposure. In addition, we describe our sample optimization efforts that led to a substantial improvement in data quality.

## Results

**Datasets were collected and processed in a controlled manner.** The three datasets used in this study are presented in Fig. 1. The samples were active state class B1 GPCR complexes: PACAP38:PAC1R:Gs (PAC1R)[22], Taspoglutide:GLP-1R:Gs (GLP-1R-TAS)[23] and GLP-1:GLP-1R:Gs (GLP-1R-GLP-1)[24]. Reconstructions from the complete set of micrographs in each dataset reached global resolutions of 2.7, 2.5, and 2.1 Å, respectively (Fig. 1a, d, g). Each dataset comprised two subsets of micrographs collected with a modification of a single experimental parameter (Fig. 1 middle and right columns, and Table 1).

The acquisition of the PAC1R dataset was started with the VPP (Fig. 1b). Approximately halfway through the data acquisition session, the phase plate was retracted and the acquisition continued with the conventional defocus method (Fig. 1c). The VPP provided a significant contrast improvement that simplified the visual identification of individual protein molecules in the micrographs (Fig. 1b). The GLP-1R-TAS dataset consisted of an initial subset acquired with zero-loss energy filtering (ZLF) (Fig. 1e) followed by a subset without energy filtering (Fig. 1f). Energy filtering noticeably improved the contrast (Fig. 1e, f). The GLP-1R-GLP-1 dataset used a 100 μm OLA for the first half of the micrographs (Fig. 1h) and no aperture for the rest (Fig. 1i). The aperture had no discernable effect on the contrast of micrographs but it blocked high-angle crystal reflections by the gold support film that appeared as bright spots in the images without an aperture (Fig. 1i, arrow). In addition to parameters that were modified during the experiment, we also investigated the effects of defocus magnitude (DEF) and total exposure (EXP) by dividing the GLP-1R-GLP-1 dataset into corresponding subsets.

All subsets were processed independently through the complete single particle workflow in Relion[25]. In the preprint version of this work[26], the data were processed with older versions of the analysis software and with optimized workflows for each subset. Furthermore, the OLA and defocus magnitude subsets were not processed independently but were derived from the final particle set after the processing of the complete GLP-1R-GLP-1 dataset. Here, we present results from processing with the latest software versions and using an identical simplified workflow for all subsets. The number of micrographs in comparative subsets was matched to ensure equal data volume. This and the more neutral processing workflow enhanced the performance differences and provided a clearer picture of the experimental parameter influence.

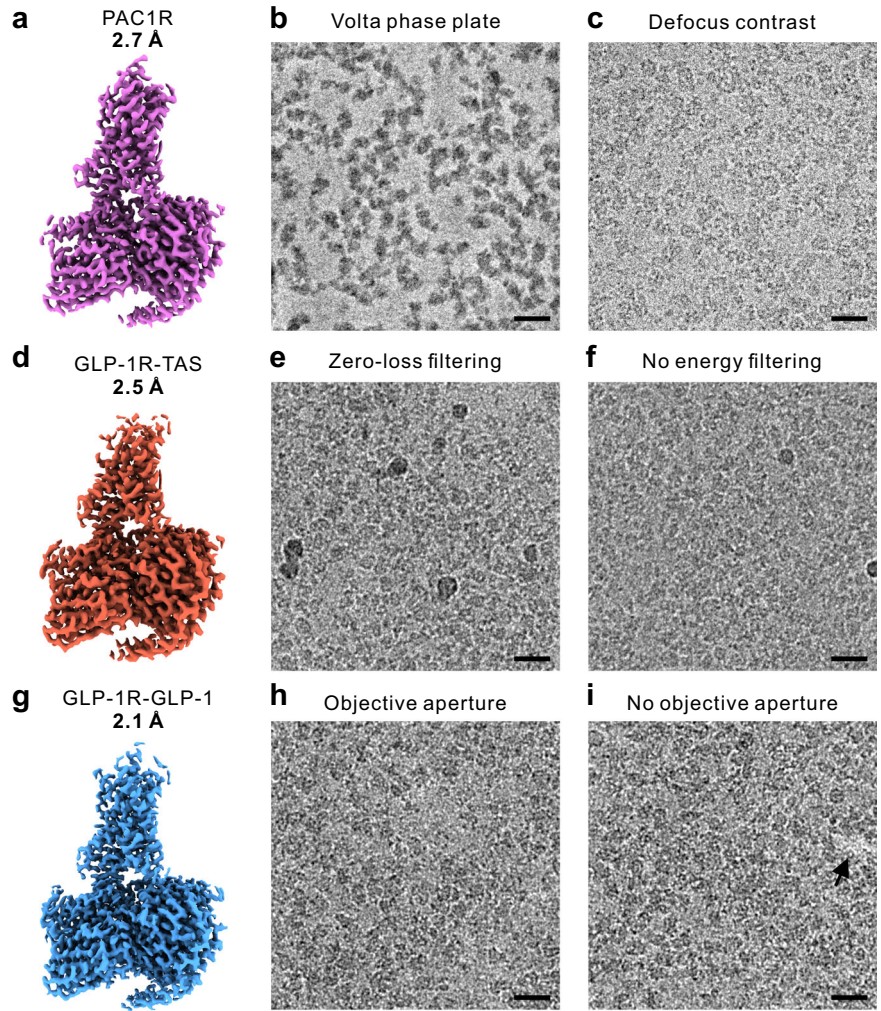

**Fig. 1 3D maps and representative micrographs from class B GPCR datasets used for the evaluation of experimental parameters. a** PACAP38:PAC1R: Gs (PAC1R) dataset collected in part with the Volta phase plate (4032 micrographs) (**b**), and in part with defocus phase contrast (3617 micrographs) (**c**). **d** Taspoglutide:GLP-1R:Gs (GLP-1R-TAS) dataset acquired partially with zero-loss energy filtering (5508 micrographs) (**e**), and partially without energy filtering (3251 micrographs) (**f**). **g** GLP-1:GLP-1R:Gs (GLP-1R-GLP-1) dataset acquired in part with a 100 µm objective lens aperture (3003 micrographs) (**h**), and in part without an aperture (2736 micrographs) (**i**). Scale bars 20 nm.

For each subset, we calculated several quantitative parameters (Table 2). Global 3D map resolution remains the most popular performance indicator in cryo-EM structural studies. However, it provides little information about the behavior of the data. The B-factor is a more comprehensive measure of the overall performance. It models the combined effect of performance-reducing factors related to the sample, the data collection, and the data processing, as a Gaussian dampening function in reciprocal space[27]. Higher values indicate stronger dampening and therefore lower performance. To determine the B-factor, the squared reciprocal resolution of independent reconstructions from random particle subsets of varying size is plotted as a function of the logarithm of the number of particles in the subsets. The B-factor is then equal to twice the reciprocal slope of the linear fit through the data points[27]. We measured the B-factors of all subsets (Table 2 and Supplementary Fig. 1) and calculated the effect of each experimental parameter on the B-factor (Fig. 2b). While the slope of the B-factor plot represents the information decay as a function of spatial frequency, the offset of the curve corresponds to the overall signal-to-noise ratio in the data. To compare this quantity, we calculated two additional performance measures from the linear fits—the resolution from 100 k particles and the number of particles necessary to reach 3 Å resolution

(Table 2, Fig. 2a, c and Supplementary Fig. 2c, d). These two values are not independent and are mathematically related to each other through the B-factor, but they give two different viewpoints on the performance and therefore we included both in the analyses. Additionally, we calculated the median value of the number of significant samples (rlnNrOfSignificantSamples) and compared the angular (rlnAccuracyRotations) and translational (rlnAccuracyTranslationsAngst) accuracies from 3D auto refinement in Relion (Table 2, Supplementary Figs. 2e, f and 3). These metrics quantify how precisely the particles align during reconstruction with lower values indicating better performance.

**The Volta phase plate reduced data quality.** Besides improved contrast, the VPP did not provide any performance benefits. Conversely, the VPP map had ~0.4 Å lower resolution than the defocus map (Fig. 2a, Supplementary Figs. 2a, c and 4a). The VPP increased the B-factor by ~40% (Fig. 2b, Supplementary Figs. 1a and 2b), and needed more than eight times more particles to reach 3 Å resolution (Fig. 2c, Supplementary Fig. 2d). Curiously, the reported angular accuracy for the phase plate data was slightly better (Supplementary Fig. 2e), but the rest of the alignment statistics favored the defocus data (Supplementary Figs. 2f and 3a).

**Table 1 Cryo-EM experiment details.**

|  | PAC1R | | GLP-1R-TAS | | GLP-1R-GLP-1 | |
| --- | --- | --- | --- | --- | --- | --- |
|  | +VPP | −VPP | +ZLF | −ZLF | +OLA | −OLA |
| Sample preparation | | | | | | |
| Concentration [mg/ml] | 5.1 | | 4.6 | | 3.7 | |
| Sample volume [μl] | 3 | | 3 | | 3 | |
| Grid type | Quantifoil R1.2/1.3 Cu/ Rh 200 | | Quantifoil R1.2/1.3 Cu/ Rh 200 | | UltrAuFoil R1.2/ 1.3 Au 300 | |
| Glow discharge time [s] | 30 | | 90 | | 90 | |
| Glow discharge current [mA] | 10 | | 10 | | 10 | |
| Blotting chamber temperature [°C] | 4 | | 4 | | 4 | |
| Blotting chamber humidity [%] | 100 | | 100 | | 100 | |
| Blot time [s] | 10 | | 10 | | 10 | |
| Microscopy | | | | | | |
| Voltage | 300 | | 300 | | 300 | |
| TEM mode | EFTEM Nanoprobe | | EFTEM Nanoprobe | | EFTEM Nanoprobe | |
| Indicated magnification | 105,000 | | 105,000 | | 105,000 | |
| C2 aperture [μm] | 50 | | 50 | | 50 | |
| Spot size | 4 | | 4 | | 4 | |
| Beam diameter [μm] | 1.55 | | 1.41 | | 1.45 | |
| Objective lens aperture/phase plate [μm] | VPP | None | 100 | | 100 | None |
| Target defocus [μm] | 0.4 | 0.7–1.2 | 0.8–1.5 | | 0.6–1.4 | |
| Zero-loss slit width [eV] | 25 | | 25 | None | 25 | |
| Pixel size [Å] | 0.83 | | 0.83 | | 0.83 | |
| Super resolution | No | | No | | No | |
| Exposure rate [e/pix/s] | 11.75 | | 15.3 | | 15 | |
| Exposure rate [e/Å$^2$/s] | 17.2 | | 22.2 | | 21.8 | |
| Exposure time [s] | 3.72 | | 3.0 | | 3.0 | |
| Total exposure [e/Å$^2$] | 64.0 | | 66.5 | | 65.4 | |
| Movie frames | 62 | | 75 | | 75 | |
| Exposure per frame [e/Å$^2$/frame] | 1.03 | | 0.89 | | 0.87 | |

The VPP had the strongest impact on performance among all tested parameters, but unfortunately it was detrimental by all indicators. This observation corroborates recent measurements of the attenuation of high-resolution signals by the VPP[28]. The signal loss is stronger than the expected loss due to electron scattering (observable in Supplementary Fig. 5a) but the actual cause of the additional attenuation remains unknown.

**Zero-loss energy filtering notably improved the performance.** ZLF had a distinctly positive effect on the performance. It improved the resolution by ~0.25 Å (Fig. 2a, Supplementary Figs. 1b, 2a, c and 4b), the B-factor by ~15% (Fig. 2b, Supplementary Figs. 1b and 2b) and reduced the number of particles required to reach 3 Å to approximately one third (Fig. 2c, Supplementary Fig. 2d). One of the primary effects of ZLF is removal of inelastically scattered electrons that do not contribute to the image contrast and add background noise[29]. However, in this case the average sample thickness was only ~17.5 nm (Supplementary Fig. 5d) and the difference between the average image intensity with/without ZLF is ~1.7 counts/pixel (~5%) (Supplementary Fig. 5c). Therefore, background noise reduction alone cannot explain the observed performance gain. Most likely, it is due to additional amplitude contrast signal generated by ZLF[30]. Such contribution is also supported by the perceptibly better contrast in ZLF images (Fig. 1e versus f). Our results recapitulate the improvement observed in a recent sub-2 Å cryo-EM reconstruction of a homopentameric GABA$_A$ receptor[6].

**The objective lens aperture showed negligible effects.** The OLA had a very small impact on the performance. The resolution values were within 0.05 Å, the B-factors were within 5% and the

number of particles necessary to reach 3 Å were within 20% (Table 2, Fig. 2, Supplementary Figs. 1c, 2 and 4c). The aperture did slightly improve the alignment performance of the particles (Table 2, Supplementary Figs. 2e, f and 3c) and therefore could be considered beneficial. Its effect was also clearly visualized by the electron count histograms of the subsets (Supplementary Fig. 5e) where it reduced the average intensity by ~0.5 counts/pixel (~1.4%), confirming that it did intercept high-angle scattered electrons.

**Larger defocus required heavier processing.** Defocus is the main phase contrast mechanism in cryo-EM. It is a deliberately introduced aberration that delocalizes and phase shifts signals proportionally to their spatial frequency[29]. When such signals are mixed with the primary unscattered wave, they create intensity modulation which can be detected by the camera. Higher defocus values cause broader delocalization and visualize larger periodicity signals, thereby increasing the overall contrast. While higher defocus makes particles easier to see by eye, during processing, strongly delocalized high-resolution signals require more care and are more difficult to recover[31,32]. Our experience showed that using lower defocus values, and the associated lower image contrast, did not impair the localization and alignment of the particles during reconstruction. On the contrary, reconstructions from datasets collected with lower defocus produced higher resolution reconstructions. To test the practical impact of defocus, we split the GLP-1R-GLP-1 dataset into two equal halves and independently processed them. The split point happened to be very close to 1 μm defocus (0.994 μm). The low defocus subset produced a slightly higher resolution reconstruction (2.36 vs 2.41 Å), had a lower B-factor (87.3 vs 94.3 Å$^2$) but also a slightly lower resolution from 100 k particles (2.64 vs 2.56 Å) and required

**Table 2 Data processing details.**

| | PAC1R | | GLP-1R-TAS | | GLP-1R-GLP-1 | | GLP-1R-GLP-1 Def. >1 μm | | Def. <1 μm | | GLP-1R-GLP-1 | |
|---|---|---|---|---|---|---|---|---|---|---|---|---|
| | +VPP | −VPP | +ZLF | −ZLF | +OLA | −OLA | 212 Å box | 332 Å box | 212 Å box | 332 Å box | Exp. 65 e/Å² | Exp. 40 e/Å² |
| Micrographs | 4032 | 3617 | 5508 | 3251 | 3003 | 2736 | 3013 | | 2726 | | 5739 | 5739 |
| Micrographs after CTF fits (retention) | 3676 (91%) | 3466 (96%) | 4631 (84%) | 2487 (76%) | 2098 (70%) | 2477 (91%) | 2288 (76%) | | 2288 (84%) | | 4575 (80%) | 4742 (83%) |
| Micrographs used | 3466 | 3466 | 2487 | 2487 | 2098 | 2098 | 2288 | | 2288 | | 4575 | 4742 |
| Measured defocus [μm] | 0.4–1.2 | 0.7–1.7 | 0.6–1.8 | 0.6–1.8 | 0.6–1.45 | 0.6–1.45 | 1.0–1.45 | | 0.6–1.0 | | 0.6–1.45 | 0.6–1.45 |
| Picked particles [×10³] | 2387 | 2791 | 1641 | 1618 | 1491 | 1489 | 1480 | | 1772 | | 3168 | 3429 |
| (per micrograph) | (689) | (805) | (660) | (651) | (711) | (710) | (647) | | (774) | | (692) | (723) |
| Final particle set [×10³] (retention) | 479 (20%) | 793 (28%) | 248 (15.1%) | 278 (17.2%) | 306 (21%) | 337 (23%) | 234 (16%) | | 385 (22%) | | 524 (17%) | 733 (21%) |
| (per micrograph) | (138) | (229) | (100) | (112) | (146) | (161) | (102) | | (168) | | (115) | (155) |
| Resolution after CTF refinement [Å] | 3.43 | 2.95 | 2.76 | 3.08 | 2.62 | 2.66 | 2.56 | | 2.59 | | 2.47 | 2.62 |
| **Resolution after polishing [Å]** | **3.22** | **2.80** | **2.62** | **2.87** | **2.44** | **2.39** | **2.41** | **2.37** | **2.36** | **2.37** | **2.21** | **2.39** |
| **B-factor after polishing [Å²]** | **188.0 (5.1)** | **135.5 (6.5)** | **113.6 (3.9)** | **134.7 (7.4)** | **91.2 (2.6)** | **87.1 (2.5)** | **94.3 (2.5)** | **88.2 (2.8)** | **87.3 (1.9)** | **83.9 (2.0)** | **80.5 (2.2)** | **94.2 (1.9)** |
| Particles to reach 3 Å after polishing [×10³] | 1787 (539) | 191 (102) | 33.8 (12) | 118.7 (70) | 19.1 (5.7) | 23.6 (7.0) | 13.9 (3.8) | 13.3 (4.2) | 24.2 (5.6) | 28.5 (7.5) | 14.9 (4.4) | 33.2 (7.3) |
| Resolution from 100k particles after polishing [Å] | 3.53 (0.07) | 3.14 (0.12) | 2.77 (0.07) | 3.03 (0.12) | 2.60 (0.06) | 2.63 (0.06) | 2.56 (0.05) | 2.52 (0.06) | 2.64 (0.05) | 2.66 (0.06) | 2.51 (0.06) | 2.73 (0.05) |
| Median number of significant samples (rlnNrOfSignificantSamples) | 35 | 29 | 24 | 31 | 19 | 23 | 19 | 24 | 17 | 28 | 18 | 20 |
| Angular accuracy [deg] (rlnAccuracyRotations) | 1.09 | 1.17 | 0.96 | 1.02 | 0.88 | 0.89 | 0.83 | 0.53 | 0.95 | 0.54 | 0.85 | 0.92 |
| Translational accuracy [Å] (rlnAccuracyTranslationsAngst) | 0.58 | 0.50 | 0.38 | 0.49 | 0.36 | 0.39 | 0.32 | 0.20 | 0.39 | 0.27 | 0.34 | 0.40 |

Unless the row title indicates otherwise, the value in brackets is the estimated standard error.
The final map resolution and B-factor of each subset are highlighted in bold.

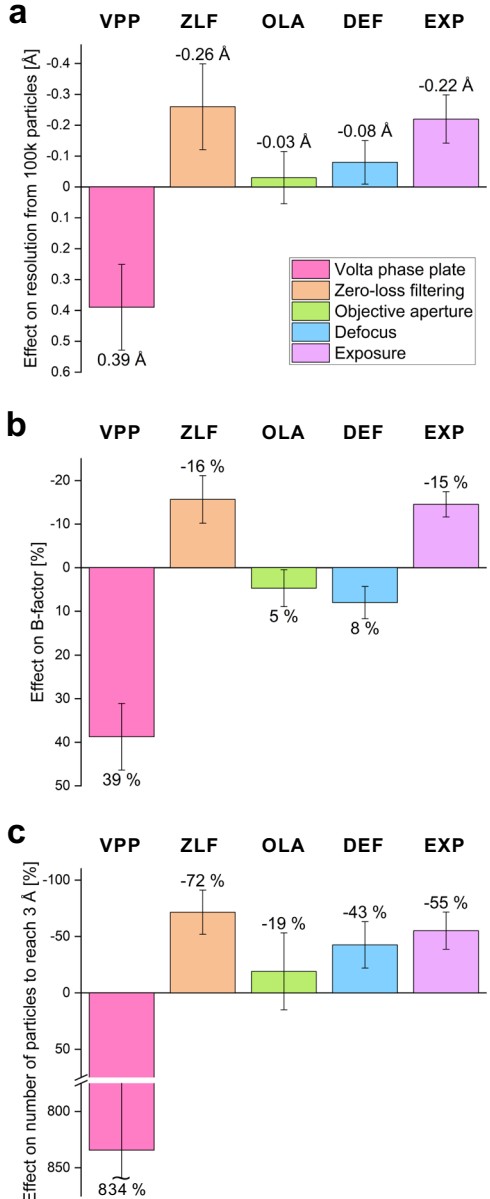

**Fig. 2 Effect of the experimental parameters on the three main performance measures.** The graphs summarize the performance effects of the Volta phase plate (VPP), zero-loss filtering (ZLF), objective lens aperture (OLA), higher defocus (DEF), and higher exposure (EXP). In all plots, an upward bar indicates an improvement in performance. **a** Effect of the experimental parameters on the resolution from 100 k particles expressed as the difference in Å (from Table 2). **b** Effect on the B-factor expressed as the change in % from the B-factor without the device, low DEF or low EXP to the B-factor with the device, high DEF or high EXP (from Table 2). **c** Effect on the number of particles to reach 3 Å resolution expressed as the change in % from the number of particles without the device, low DEF or low EXP to the number of particles with the device, high DEF or high EXP (from Table 2). Error bars represent the standard error of each value estimated from the B-factor linear fits through $7 \leq n \leq 9$ independent 3D reconstructions from random particle subsets of varying size (Supplementary Fig. 1a–e).

~40% more particles to reach 3 Å (Table 2, Fig. 2, Supplementary Figs. 1d and 2). Overall, the lower defocus subset did not show significant performance deficits, despite the lower contrast in the images. Nevertheless, alignment accuracies, sharpening B-factor, and possibly particle picking and classification were affected by the lack of low frequency components (Table 2, Supplementary Figs. 2e, f and 4d). For these results we used a particle box size of 212 Å that was 1.5 times larger than the particle size (~140 Å) and could accommodate delocalized signals from the center of the particle at the final resolution (2.4 Å) and defocus of 1.3 μm. In regions away from the center of the box, however, there may be loss of high-resolution information due to signal delocalization. To test if this was the case, we re-polished the particles with a larger 332 Å box, that was ~2.5 times the particle diameter, which should accommodate delocalized signals from all well resolved portions of the complex. The B-factor plot indeed showed an improvement of the high defocus relative to the low defocus line (Supplementary Fig. 2d). The global resolutions of the two subsets also matched at 2.37 Å, the B-factor difference decreased to ~5% (Supplementary Figs. 1d and 4f), and the number of significant samples distributions changed in favor of the higher defocus (Supplementary Figs. 3d, f).

**Higher exposure is advantageous for small particles**. The final experimental parameter that was tested was the total exposure. There is no general consensus about the exposure threshold past which there will be no further gain in the quality of reconstructions. It will depend on the amount of signal in each projection, hence the size of the particle, with smaller molecules possibly benefiting more from higher exposures[33]. The direct detector movie acquisition format combined with exposure weighting and Bayesian polishing provides optimal signal extraction from the frame stack[34]. In principle, this means that images can be exposed indefinitely. However, longer exposures reduce the acquisition, data storage and data processing throughputs and a practical compromise is necessary. Nowadays, the exposure used for single particle analysis is typically in the range of 30–80 e/Å². To test the effect of total exposure on the performance for GPCRs, we re-processed the GLP-1R-GLP-1 dataset, using a 40 e/Å² (46 frames) subset from the original 65 e/Å² (75 frames) movies. The results showed a decrease in performance with lower exposure (Table 2, Fig. 2, Supplementary Figs. 1e and 2). The 40 e/Å² subset produced a ~0.2 Å lower resolution from 100 K particles and needed approximately twice the number of particles to reach 3 Å (Table 2, Fig. 2). This indicates a noticeable benefit from using higher total exposures (>60 e/Å²) for GPCRs. It is not clear why the increase from 40 to 65 e/Å² had a positive effect at high resolutions. In the wake of radiation damage, only low-resolution features of the sample (>7 Å) would be contributing to frames beyond 40 e/Å² [33]. For the same reason, we expected that low defocus particles, that already have weak overall contrast, will be affected more by limiting the exposure. There was indeed a small decrease in the lower defocus particles fraction in the final particle set of the 40 e/Å² subset (Supplementary Fig. 1f). The B-factor of the 40 e/Å² subset was also noticeably higher (94.2 vs 80.5 Å²) (Supplementary Fig. 1e).

**Gold foil grids drastically improved the performance**. Unsurprisingly, the quality of frozen grids had a significant impact on the outcome of GPCR experiments. High purity, good homogeneity and sufficient concentration of the protein solution were essential for achieving good results. To this end, all GPCR protein preparations were characterized for purity, assembly state and stability by biochemical assays and negative stain electron microscopy[8,22,23]. During the initial experiments, we screened the cryo-EM grid plunging conditions, such as glow discharge time, blot force and blot time for our combination of plasma cleaner and plunging device. After determining a set of parameters that consistently produced thin ice over large portions of the grid, we

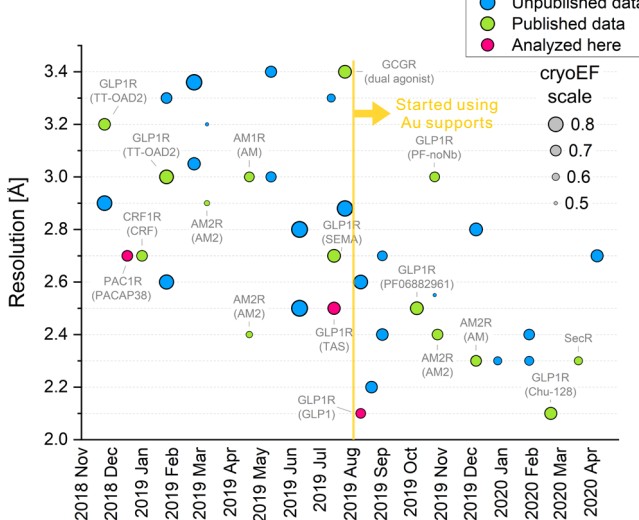

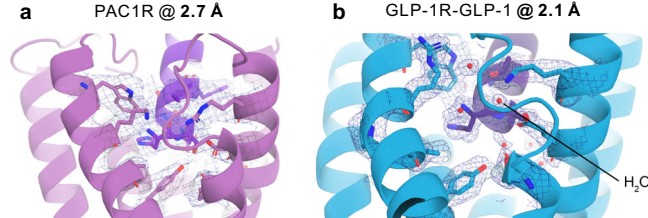

**Fig. 4 Illustration of the atomic modeling benefits from improved map resolution.** Cartoon representation of the N-terminal portion of the agonist peptides PACAP38 and GLP-1 bound to their respective receptors. The cryo-EM density maps are drawn at 5-sigma as a blue mesh and highlight the ability to more accurately model side chain and water positions at a higher spatial resolution. **a** N terminus of the PACAP38 peptide (purple) bound to the PAC1 receptor (pink) at 2.7 Å resolution. There were no reliably detectable water molecule densities in this region of the map. **b** N terminus of the GLP-1 peptide (dark purple) bound to GLP-1 receptor (light blue) at 2.1 Å resolution. Several water molecule densities were identified and modeled inside the binding pocket of the receptor where they facilitate the interaction with the ligand.

**Fig. 3 Resolution history of our cryo-EM GPCR reconstructions.** Over the year and a half period shown in the plot, there is a general trend towards better resolution with a significant improvement in August 2019, when the sample supports were switched from holey carbon films to gold foil grids. The size of each dot represents the cryo-EM efficiency quantity (cryoEF, see values in the legend) that estimates the uniformity of the particle orientation distribution. A value of ~0.5 represents an angular distribution that is barely sufficient for producing a usable 3D map and a value of 1 corresponds to an ideal uniform distribution. Published results are annotated and the three datasets analyzed in this study are highlighted in pink.

kept these parameters constant. We had to significantly extend the glow discharge time to 90 s (Table 1), to improve grid wettability and enhance sample drainage during blotting. This reduced the pooling of solution in the middle of grid squares, especially with 200 mesh grids (compare Supplementary Fig. 7a and b with 7c and d). We also settled on a relatively long blotting time of 10 s that produced uniformly thin ice more consistently (Table 1). For each new sample, we only screened the concentration by preparing 2–3 grids with 2× dilution in-between. In our experience, GPCR samples produce optimal grids at concentrations between 3 and 7 mg/ml. The highest resolution results came from grids with uniformly thin ice that contained a single layer of molecules covering 50–90% of the image area (Fig. 1, Supplementary Fig. 7).

Figure 3 shows the resolution history of our GPCR observations in a 17-month period from November 2018 until April 2020. There is a general trend towards better resolution with a pronounced jump in August 2019. At that point, we started using gold foil support grids[35] instead of the holey carbon grids that were used in all previous experiments (Table 1). This improved the resolution substantially, by ~0.5 Å. Previously, we had achieved 1.62 Å resolution with an apoferritin test sample on the same microscope by using holey carbon grids[1]. Therefore, we did not suspect that carbon film grids were imposing a performance penalty on GPCR samples, where the resolution was typically in the 2.5–3.0 Å range (Fig. 3). Nevertheless, the improvement from the gold foil grids was evident and was confirmed consistently by the follow-up experiments (Fig. 3). Quantitative analysis of the ice thickness in the GLP-1R-TAS and GLP-1R-GLP-1 datasets showed that it was very similar and in the range of 150–200 Å (Supplementary Fig. 5d, f). Therefore, the resolution improvement was not due to thinner ice. The PAC1R dataset had a much wider thickness distribution that extended towards thicker ice (Supplementary Fig. 5b), which could explain

its slightly lower resolution. Beam-induced motion statistics showed a significant reduction in the early and total displacements when using a gold foil grid (Supplementary Fig. 6). The early stage of an exposure carries the highest resolution signal because of low accumulated radiation damage. Together with the expected vertical doming of the ice layer on carbon film supports[36], this could explain the strong benefit of using gold foil grids. While the performance still depends primarily on sample quality, since switching to gold foil grids we are routinely obtaining resolutions at or below 2.5 Å (Fig. 3) and in some cases below 2 Å[37].

**Orientation uniformity did not correlate with resolution.** To further characterize the behavior of samples, we calculated the cryo-EM "efficiency" (cryoEF)[38] for all datasets (Fig. 3, dot size). CryoEF measures the uniformity of the particle orientation distribution and values above 0.5 are considered to indicate that a dataset will produce a usable 3D map. All datasets in Fig. 3 satisfied this criterion, with only four of them having cryoEF values between 0.5 and 0.6 and the majority in the range 0.65 to 0.75. There was no strong correlation between cryoEF and map resolution. The transition to gold foil grids appears to have truncated high (>0.75) cryoEF values without a negative impact on resolution, possibly because of consistently thinner ice that excludes particles oriented with their long axis perpendicular to the ice layer.

**Map fidelity is important for identification of waters.** Figure 4 shows a similar region from the PAC1R and the GLP-1R-GLP-1 maps and models to illustrate the increase in map fidelity from 2.7 to 2.1 Å. The most significant improvement is in the definition of side chain densities, which increases the probability of correct rotamer orientation assignment, the accuracy of interaction distance measurements and the identification of water molecules (Fig. 4b). Such information can be crucial for the correct interpretation of the interactions between the receptor and the ligand, which could be a peptide or a small molecule.

## Discussion
The systematic testing of single experimental parameters revealed two major and several minor factors that influence data quality. The first major observation was that the VPP does not provide any benefits in GPCR cryo-EM studies and therefore researchers

should refrain from using it in their experiments. Except for grid type, the rest of the parameters had much smaller but nonetheless important contributions to the performance.

Zero-loss energy filtering improved the resolution by ~0.25 Å. Here, we used an energy selection slit width of 25 eV due to long-term stability issues with our hardware setup. Recent sub-2 Å results of a GABA$_A$ receptor acquired with a much narrower 5 eV slit demonstrated similar benefits[6]. The contribution of energy filtering appears to be due to additional amplitude contrast generated by removing inelastically scattered electrons. However, the improvement of high-resolution components may indicate that there could be other effects that will need further investigation.

The OLA had a slight positive effect on some of the performance measures, such as alignment accuracy. With the gold foil grids it also prevented crystal reflection spots from appearing in the images. Consequently, we consider the aperture to be beneficial, but it should be used with care because it may cause wavefront distortion of high-resolution components (<2 Å) due to electrostatic potentials on the aperture, as reported recently[6].

Our results show that there is a practical advantage of including low defocus values (<1 μm) during data collection from optimized samples. In our GPCR experiments, we typically use a defocus range of 0.5–1.5 μm. Current particle handling strategies in the most popular image processing packages impose a restriction on the minimum particle box size that will preserve delocalized high-resolution components at the maximum defocus in the dataset. As demonstrated here, higher defocus values necessitate larger box sizes that could be impractical and/or reduce productivity. Similar to processing approaches from the early days of cryo-EM[39], the recently introduced M package overcomes this limitation by performing a CTF sign correction on a larger area before extracting the particle box[32]. This was shown to solve the resolution limitation of higher defocus values and could allow their use in the future without detrimental effects.

We measured a ~0.2 Å resolution benefit from using a higher 65 e/Å$^2$ total exposure compared to 40 e/Å$^2$. The merit of higher exposure is most likely due to stronger low frequency components that improve the effectiveness of particle picking and classification steps during data processing.

Sample optimization had a significant impact on the quality of results. High protein concentrations, to get a monolayer of molecules in thin ice, improved the quality and consistency of the grids. Gold foil supports had the largest positive impact among all tested parameters. They improved the resolution of reconstructions by ~0.5 Å. Their advantage appears to be a combination of diminished beam-induced motion and more consistent sample quality.

The pixel size and exposure per frame are also important experimental parameters. We do not have comparative data for them, but we usually aim for a pixel size that is closest to one third of the target resolution, e.g. 2.5 Å/3 = 0.83 Å. This places the target resolution at two thirds of the physical Nyquist frequency of the detector. With densely packed smaller particles, it may be beneficial to go to a smaller pixel size that is a quarter of the target resolution, placing it at half of the physical Nyquist periodicity. For the exposure per frame, we aim for ~0.8 e/Å$^2$/frame.

Other factors, that are outside of the scope of this study, will also have a substantial influence on the performance. The size and flexibility of a protein complex are fundamental attributes for cryo-EM. Smaller and/or more dynamic molecules will generally produce lower resolution reconstructions. Recent examples are the PF 06882961-GLP-1R-DNGs[40] (shown in Fig. 3 as GLP-1R (PF-noNb)) that did not include the proprietary stabilizing nanobody 35, and two CGRP receptor-only structures[41] that were less than half the size of a ternary GPCR-G protein complex.

For such challenging targets to become routine and improve in resolution, basic cryo-EM technologies must be advanced further. In this respect, direct detectors continue to evolve, and recent innovations include an electron event representation format that mitigates the limitation of fixed movie frame time by recording every individual electron event[42]. This and other developments will contribute to achieving better signal-to-noise ratio and higher throughput that will provide significant benefits for difficult samples.

The experimental parameter optimization presented here can be applied to other similarly sized membrane proteins and should result in comparable performance gains. Nevertheless, our results and conclusions are based on highly-optimized GPCR samples and sample quality issues can easily override the benefits from experiment fine-tuning. Therefore, sample quality optimization should remain the top priority in any cryo-EM project, before any subsequent experiment parameter adjustments.

## Methods

**Protein expression and purification.** The expression and purification of the PAC1R[22], GLP-1R-TAS[23] and the GLP-1R-GLP-1[24] was described in detail previously.

**Cryo-EM sample preparation.** Sample support grids were washed in advance by placing them on a piece of filter paper in a glass Petri dish, soaking the paper with acetone and letting the solvent evaporate. Immediately before sample preparation, the grids were glow discharged in low pressure air with 10 mA current in a PIB-10 Ion Bombarder (JEOL, Japan). Cryo-EM grids were prepared by plunge-freezing in liquid ethane on a Vitrobot Mark IV (Thermo Fisher Scientific, USA). The glow discharge and plunge-freeze parameters are listed in Table 1.

**Cryo-EM data collection.** The datasets were collected on a Titan Krios G3i (Thermo Fisher Scientific, USA) 300 kV electron microscope equipped with a GIF Quantum energy filter and a K3 direct electron detector (Gatan, USA). Movies were acquired with homemade scripts in SerialEM[43] employing a 9-hole beam-image shift acquisition pattern with 1 image in the center of each hole and saved as non-gain-normalized compressed TIFF files. The acquisition parameters for each dataset are listed in Table 1 and the dataset details in Table 2.

**Data processing.** The processing workflow trees for the investigated subsets are presented in Supplementary Figs. 8–12. All datasets were motion corrected with MotionCor2 v1.4.2[44] using 5 × 3 patches (long × short edge of the K3 image area), no frame grouping, B-factor 500, with saving of dose weighted and non-dose weighted averages. The CTFs were fitted on the non-dose-weighted averages using Gctf v1.18b2[45] with 20–3.5 Å resolution range, 1024 pixel box, EPA averaging, high-resolution refinement with 15–3.0 Å resolution range, resolution cross-correlation cutoff limit of 0.5 and defocus search range determined from the defocus histogram of an initial trial CTF fit. A 20 Å low-pass filtered GPCR map from a previous reconstruction was used for reference-based initial particle picking of 1000 random micrographs from each subset followed by a round of 2D classification and an initial 3D refinement with a spherical mask in Relion 3.1.2[25]. The resulting 3D map was used as a reference for reference-based picking of all micrographs and to create a 3D mask for the initial 3D classification rounds. All subsets were cleaned solely by 3D classification, which in our opinion is less-prone to operator bias and provides more consistent results. The initial particle sets (1.66 Å/pix, box 128 pix) were subjected to two rounds of 3D classification with 3 classes and halving of the angular and translational search parameters between the rounds. We used default regularization parameter (tau fudge) values of $T = 2$ for 2D and $T = 4$ for 3D classifications. A 20 Å low-pass filtered internal reference and "fast subsets" (small random particle subsets in the initial iterations) were used to prevent bias propagation and to provide featureless references for false-positives and contaminants to be classified into. Particles from the best resolved class of the initial 3D classification were subjected to three 3D auto-refinement runs with re-extraction with a smaller pixel (1.21 Å/pix, box 176 pix) and CTF refinement between the runs. The particle stack was then further cleaned through a 3D classification with 5 classes and small local searches (same search steps as in the last iteration of the previous 3D auto-refine), which in our experience works better than no-alignment 3D classification for sorting-out low-quality particles. Particles from the best resolved class were subjected to two 3D auto-refinement runs with CTF refinement in-between followed by Bayesian polishing and another two 3D auto-refinements with CTF refinement in-between. For the GLP-1R-GLP-1 subsets, the pixel size was reduced (1.0 Å/pix, box 212 pix) during the Bayesian polishing step because the reconstructions were reaching Nyquist periodicity. For the defocus magnitude subsets, the Bayesian polishing was repeated also with a larger box (1.0 Å/pix, box 332 pix) followed by the same two rounds of 3D auto-refinement with a

CTF refinement in-between as for the smaller box, to evaluate the box size effect on the performance of higher defocus. The 40 e/Å$^2$ exposure frame subset of GLP-1R-GLP-1 used the first 46 of the 75 movie frames in the motion correction and Bayesian polishing steps. All post-processing steps used the same mask as the corresponding 3D auto-refinement step. Masks were prepared based on the 3D map from the preceding 3D auto-refinement or 3D classification step using a threshold that either included or excluded the micelle. The mask parameters and the inclusion of the micelle are indicated in the processing workflows (Supplementary Figs. 8–12).

For all subsets, after the last 3D auto-refinement step, B-factor estimation random particle subset refinements were performed using the "bfactor_plot.py" script supplied with Relion. Low particle number outlier points (<1–2 k particles), where the 3D refinement did not lock onto the alpha helical pitch, were excluded from the B-factor linear fits (gray points in Supplementary Fig. 1). The standard errors of the values in Table 2, Fig. 2 and Supplementary Fig. 2 were estimated from the standard errors of the slope and offset values of the B-factor linear fits in Supplementary Fig. 1.

The cryo-EM efficiency (cryoEF) value was calculated on the particle set from the final 3D refinement of each dataset using the cryoEF program[38] with 256 pixels box size with the commands: "extractAngles.sh run_data.star > angles.dat", where the "run_data.star" file is the particle star file from the final 3D auto-refinement; and "cryoEF -f angles.dat -b 256 -B Bfac -D 132 -r Res", where the *Bfac* and *Res* parameters are the sharpening B-factor and resolution from the final post-processing job in Relion. The ice thickness in Supplementary Fig. 5 was calculated using the formula $t = \text{MFP} \cdot \ln(I_0/I)$, where MFP is the inelastic mean-free path, $I_0$ is the open beam intensity measured from images that were taken in holes without ice, and $I$ is the average intensity of a micrograph. The inelastic mean-free path values used in the formula were 395 nm without an objective aperture and 322 nm with an aperture[46]. In the preprint version of the manuscript[26] we used the early and total motion values reported by Relion, which represent the total motion distance calculated as the sum of the absolute shifts between frames. However, when the beam-induced movements are small, and the number of frames is large, the motion distance values tend to be dominated by the noise in the inter-frame shifts. Therefore, here we chose to use the displacement between the first frame and the frame corresponding to ~4 e/Å$^2$ exposure (frame No. 5 for the PAC1R and frame No. 6 for the GLP-1R-TAS and GLP-1R-GLP-1) for the early displacement and between the first and the last frame for the total displacement. These statistics may downplay non-monotonous (e.g. U-turn) beam-induced movements, but showed a very clear motion magnitude difference between the carbon and gold foil support grids (Supplementary Fig. 6).

**Reporting summary**. Further information on research design is available in the Nature Research Reporting Summary linked to this article.

## Data availability

The data needed to evaluate the conclusions of the paper are present in the paper and/or the Supplementary Information. SerialEM scripts and additional data related to this paper may be requested from the authors. The global cryo-EM maps of PAC1R, GLP-1R-TAS, and GLP-1R-GLP-1 are deposited in the Electron Microscopy Data Bank (https://www.ebi.ac.uk/pdbe/emdb/) under accession numbers EMD-0993, EMD-22883, and EMD-21992, respectively. The previously determined atomic models of PAC1R[22], GLP-1R-TAS[23], and GLP-1R-GLP-1[24] have been deposited to the Protein Data Bank (https://www.rcsb.org/) under accession codes 6P9Y, 7KI1 [https://doi.org/10.2210/pdb7KI1/pdb], and 6 × 18, respectively. The complete PAC1R and GLP-1R-GLP-1 datasets have been deposited to the Electron Microscopy Public Image Archive (https://www.ebi.ac.uk/pdbe/emdb/empiar/) under accession codes EMPIAR-10359, and EMPIAR-10673, respectively.

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

## Acknowledgements

We thank Prof. Masahide Kikkawa for providing access to his lab's facilities, Dr. Haruaki Yanagisawa for help with data management and Kazuhiko Nakamura for administration of the cryo-EM facility. R.D. was supported by the Japan Society for the Promotion of Science (JSPS) KAKENHI #18H06043, Takeda Science Foundation 2019 Medical Research Grant and Japan Science and Technology Agency PRESTO (18069571). F.E. is a Japanese Society for the Promotion of Science (JSPS) International Research Fellow. D.W. is a Senior Research Fellow of the Australian National Health and Medical Research Council (NHMRC). P.M.S. is a Senior Principal Research Fellow of the NHMRC. The work was supported by a Program grant (1150083) and Project grants (1120919, 1126857 and 1159006) from the NHMRC. M.B. is supported by a US-DoD grant (PR180285, W81XWH-19-1-0126).

## Author contributions

R.D., D.W. and P.M.S. conceived the research. R.D. developed the method, collected and processed cryo-EM data, analyzed results and wrote the initial draft. M.B. and F.E. processed cryo-EM data and analyzed results. Y.L.L. and X.Z. expressed and purified protein complexes, performed biochemical assays, negative stain screening and analyzed cryo-EM data. All authors participated in the editing of the manuscript.

## Competing interests

The authors declare no competing interests.
