## [Peer Review File · Nature Communications]

REVIEWER COMMENTS

Reviewer #1 (Remarks to the Author):

Here, the authors provide an in-depth investigation into the influence of sample preparation and data collection parameters on the ability to routinely determine 3 Å or better resolution cryo-EM structures of GPCR specimens. The manuscript is well-written and the data are clearly presented. The systematic dissection of the contributing parameters during EM grid preparation and data collection are a worthy contribution to a rapidly-evolving field where such parameters are easily influenced by anecdotal evidence and subjective preferences. This reviewer only has a few minor criticisms, mostly pertaining to the data processing and evaluation of the final particles.

Comments/criticisms:

-Based on the evidence provided in the manuscript, particularly Figure 3, there are still several structures (post Aug 2019) that are not resolved to the 2.5 Å resolution threshold the authors reference. Although some are unpublished, GLP1R (PF-noNb) represents a published data set that was not resolved to 2.5 Å or better. I am curious if all of the optimized parameters outlined in the manuscript were used for this data set? If not, which parameters were not utilized? Perhaps there is more information to be gleaned about why this GLP1R sample did not perform as well as the others.

-Although the authors use a series of metrics to evaluate the final maps, the “alignability” of the particles was not evaluated. Specifically, parameters outputted by RELION (e.g., `rlnNRSignificantSamples`, etc.) that might further inform on the effects of data collection parameters outlined here. I find the lack of such analyses surprising given the rigor in the paper. Perhaps the authors already looked into such metrics but did not find anything conclusive worth reporting? In particular, I am curious how histograms of such values compare for a given protein across data collection parameters and if certain data collection schemes

-The number of frames used during movie recording was increased for the GLP1R data sets, compared to PAC1R, which were consistently resolved to higher resolution. Given the impact of early motion correction to the final resolution, do the authors think the shorter frames also benefited frame alignment and recovery of high-resolution information for the 3D reconstruction?

-Surprisingly, the authors did not provide any data processing trees. Although these proteins were previously published, presentation of the data processing trees would be beneficial and help inform on the quality differences at each stage of 3D classification. In particular, it would help to know if the same 3D classification parameters (i.e., tau fudge, etc.) values were used for each successive 3D classification or if they were varied? And why? Which values did the authors settle on?

- Can the authors comment on why they did not use 2D classification for pruning the dataset once template-based picking was employed? I find this a bit perplexing and am curious which step in the 3D processing benefits the removal of particles exhibiting template model bias?

-I find the lack of FSC curves in the manuscript surprising. Although some of the reported resolution values are statistically insignificant, differences in the FSC curves could be informative. Furthermore, Guinier analyses could provide even further information about the differences in quality between maps when the reported resolution values are minor.

-I wonder if the authors could comment on the magnification selected for imaging and whether this was also optimized similar to the other data collection parameters? Seems to be a perfect fit for the resolution range of most structures but curious if slightly higher mags were also tested.

-Density subtraction of the detergent micelle is a common practice for small membrane proteins but it doesn't appear the authors employed this. Has it been attempted but not beneficial?

Minor:

Fig. S4. The green lines for the polygons are very difficult to see.

Table 1: "phase palte" should be corrected

-Mark Herzik

Reviewer #2 (Remarks to the Author):

It has been a real pleasure to read the manuscript by Danev et al., as I have been closely following this collaboration for years. It has been truly pioneering work in the field of cryoEM structure determination of GPCR-G protein complexes and they have had many milestone achievements. 
While the technique of cryoEM has made remarkable contributions to biological research as a whole, ideal experimental parameters are typically very hard to find, are usually just propagated word-to-mouth, and are usually of anecdotal nature. The manuscript shows hard, experimental evidence for best practices in cryoEM, which can not only be applied to GPCRs. Besides, Prof. Danev has spent countless years on the development of the VPP and I applaud him for his honest evaluation of the contribution of the VPP on data quality.

The manuscript has been uploaded to bioRxiv in August and has been very well received by the community. The paper has been downloaded over 2000 times and it already has been cited twice, which is remarkable. The manuscript by Danev et al. will be an invaluable reference for training the next generation of electron microscopists. It can be considered an "instant classic" and I highly recommend publication with minor revisions.

The only substantial questions and remarks I have are regarding the data processing schemes. I have spent quite a bit of time looking at the supplementary text and table 2:

- From the supplementary text alone it is unclear, at what point exactly the datasets were split. A flowchart, highlighting the key differences, and at what point the datasets were split, would be helpful to keep track of things. I have attempted to assemble such a flowchart (see below). From my understanding picking was done on the 'full' datasets (except the 'low dose' dataset?). The +VPP/-VPP, low dose/high dose and +ZLF/-ZLF were processed separately after picking (3D class, ctf refinement, etc.), while +OLA/-OLA and high/low defocus were separated only at a later stage (after 3D classification, CTF refinement, polishing). This might well be a misunderstanding on my end, but I cannot really follow the reasoning behind this discrepancy and would be interested to hear the authors justification for not treating every dataset 'equally', meaning either treat all datasets completely separately, or split them at the same point. I could well imagine that some of the results shown (or the lack of a significant effect) could be influenced by this difference in treatment.

- Picking: This mostly is a continuation of my previous point. It appears that the number of picked particles per micrograph varies between the 'half' datasets (by 10-25%), and as stated above in some cases picking was done on the consensus dataset, in some cases individually on the split datasets. The difference appears to be smaller for the 'consensus treatment' (i.e. OLA and defocus), while separately treating datasets resulted in larger differences in picked particles per micrograph (VPP, ZLF and dose). This difference could very well arise from superior data quality with the beneficial experimental parameter, but it is an important caveat to mention that one might be able to compensate (or at least influence the result) by tweaking the parameters towards the 'weaker' data. It could theoretically also point to sample variation, which would introduce artifacts in the interpretation. Another complication regarding picking is that the centering for each dataset could be very different. It would be helpful to report average '_rlnOriginXAngst/_rlnOriginYAngst' after 2D classification, which reports the distance from the original center to the refined center, to show that the "quality" of the

centering was comparable.

- Defocus: The notion that low defocus values used in data collection result in better reconstructions, has gained traction in the recent past. However, a recent publication by Glaeser et al. (Ultramicroscopy) has questioned this “dogma” and suggests defocus values as high as 5 μm to be ideal. It would be worth commenting on this in the paragraph about the effect of defocus values on reconstructions. To underline the positive effect of smaller defocus values, it would be possible to report the results of more than one dataset, since all datasets have the same total range of -0.7 to -1.7 μm . There is also mention of a ‘PAC1R defocus subset’ in the supplementary text, but no further discussion about it, as far as I can tell.

- Masks: Masking is known to have a significant influence on resolution estimates. While I am sure that the authors have paid great attention to using masks with similar ‘features’, it would be worth mentioning how they were generated (B-factors, smooth edges, etc.), to ensure that the two compared datasets were treated equally.

Other minor comments:

- “Beam-induced motion statistics showed a clear reduction in the early motion with 37 the Au foil grid (Fig. S5A) but similar total motion (Fig. S5B).” This is a very interesting and unexpected finding. This implies that the positive effect of using gold grids is only with respect to movement on the first frames, but not the overall doming of the ice. It would be very interesting to plot a comparison of the per frame motion, as I would guess the motion on the gold grids would be more uniform (or something more like a gaussian?).

- ZLF: From what I understand, the energy filter alone without using the slit (i.e. ‘-ZLF’) can introduce artifacts and could potentially produce data of inferior quality compared to a system without any energy filter. This caveat would be worth mentioning in either main text or the supplementary.

- “The recently introduced M package overcomes this limitation by performing a CTF sign correction on a larger area before extracting the particle box.” It is fair to state, that most ‘classic’ data processing packages (IMAGIC etc.) used to correct for the CTF before extraction.

So long.

Cornelius Gati

Reviewer #3 (Remarks to the Author):

This manuscript describes a comparative study of using different specimen grid preparation and electron microscope settings to collect images of three different single particle specimens of G protein receptor membrane proteins. The resolution differences between maps resulting from different settings are within a fraction of \AA . The results are not surprising to experienced electron microscopists though they may be useful to novice users. One of the shortcomings of the manuscript is nothing conceptually new. But the studies were done well in each comparison. I noted that some relevant literature was missing in the current manuscript. It would have been better if the comparison was done with a single specimen with various combinations of parameter choices. The followings are my comments on each of the experimental variables used in the study.

Volta phase plate (VPP): it is known that Volta phase plate yields impressive low resolution image contrast but does not yield higher resolution. This is why not too many reported structures using VPP except the first author who invented the VPP. It has been shown in the literature that there is no practical advantage to use the VPP from the point of view of pursuing high resolution studies for small biomolecules (e.g., G. Lander). However, PP is still of meritorious value in a recent study (e.g., R.

Glaeser).

Energy filter: there was an early study to discuss the physical rationale of energy filter in affecting the image contrast (e.g., J Langmore). It is generally agreed that energy filter is useful for thick specimen or thick embedding ice. There was a recent atomic resolution structure report which was done with and without energy filter in two different electron microscopes and cameras (e.g., W. Chiu). I agree that the current study is well-executed to show the difference with and without energy filter. However, the difference in resolution is not substantial in general.

Objective aperture: It has been known that objective aperture is critical for obtaining higher quality low dose images (e.g., PNT Unwin). However, it is nice to see that there is not much difference in the current generation of instrumentation.

Defocus setting: It is well known from the phase contrast theory that closer to optimum defocus setting is the preferred choice if high resolution is pursued. What is shown agrees with expectation.

Electron dose limits retrievable high resolution structure features due to radiation damage. This result is interesting. In rationalizing it, the observed phenomenon may be attributable to the proper weighting of the movie frames. The higher dose data yields better statistically defined particle images which would improve the particle orientation estimation accuracy and thus produces a better resolution map.

The following are a few detailed suggestions on the current manuscript

1. In this study, for objective lens aperture testing, the authors chose the 100 micron. Did the authors try the 70 micron aperture? Is it because 70 micron one limits the high resolution on the scope? (page3, line28)
2. Page4, line 2. This sentence needs to be revised and clarified.
3. In Table 1, "Grid type" showed twice in the table; I am also curious why the glow discharge time is so different (30s vs 90s) on the same type of the grids (Quantifoil R1.2/1.3 Cu/Rh 200). Did the author try other types of the grids rather than R1.2/1.3, such as R2/1 or R3.5/1.
4. Page4, line 30. Why did the author choose 25 eV energy slit instead of smaller one, since they used 5 eV before and achieved a better resolution structure.
5. Page6, line 13. The authors mentioned "blot force" was one of the parameters they optimized. This value should be added in Table 1.
6. In Fig.3, it is better to mark the same sample using the same shape for better understanding.
7. Page9, Line23-26, can the author describe a little more details on how to perform the cryoEF and measure the ice thickness instead of just a citation?

The Attachment as provided by Reviewer #2 is on the following page:

PAC1R
GLP-1R-TAS
GLP-1R-GLP-1

used **old 3D model** for picking
used **old 3D model** for picking
generated **new reference** for picking

PAC1R **+VPP**
PAC1R **-VPP**

SPLIT? → Picking → **SPLIT?** → 2x 3D → 2x CTF ref → 3D ref → Polishing → 3D ref
SPLIT? → Picking → **SPLIT?** → 2x 3D → 2x CTF ref → 3D ref → Polishing → 3D ref

Question: Picking and initial 3D class on combined VPP set?
Question: Picking and initial 3D class on combined VPP set?

GLP-1R-TAS **+ZLF**
GLP-1R-TAS **-ZLF**

SPLIT? → Picking → **SPLIT?** → 3x 3D → 2x CTF ref → 3D ref → Polishing → 3D ref
SPLIT? → Picking → **SPLIT?** → 3x 3D → 2x CTF ref → 3D ref → Polishing → 3D ref

Question: Picking on combined set?
Question: Picking on combined set?

PAC1R **defocus**

3x 3D → **???**

Question: What happened to this dataset?

GLP-1R-GLP-1

→ Picking → 3x 3D → 2x CTF ref → 3D ref → Polishing → 3D ref → **SPLIT!** to **OLA** and **defocus** sets

GLP-1R-GLP-1 **"Low dose"**
GLP-1R-GLP-1 **"High dose"**

SPLIT! → Picking → 3x 3D → 2x CTF ref → 3D ref → Polishing → 3D ref
SPLIT! → Picking → 3x 3D → 2x CTF ref → 3D ref → Polishing → 3D ref

List of changes

1. Reprocessed all particle subsets independently through an identical workflow and with the latest versions of Relion (3.1.2) and MotionCor2 (1.4.2). Updated all figures and the main text with the newly calculated results.
2. Added Fabian Eisenstein as a co-author for taking part in the reprocessing of the datasets.
3. Omitted the B-factor plots of non-polished particles because they did not contribute valuable information while at the same time reduced the clarity of the figures.
4. Added subsection headings in the Results section.
5. Added Supplementary Figure 3 containing Number of Significant Samples distributions for all particle subsets.
6. Added Supplementary Figure 4 containing Gold-standard Fourier Shell Correlation (FSC) plots for all subsets.
7. Added Supplementary Figure 2 panels **e** and **f** containing Angular accuracy and Translational accuracy for all particle subsets.
8. Replaced panels **a** and **b** in Supplementary Figure 6 (previously Fig. S5) with distributions based on the early and total displacement, rather than the total motion. The displacement was a much better metric of the beam-induced motion, because it was resistant to motion noise, unlike the total motion which was dominated by it.
9. Removed panel **c** from Supplementary Figure 6 because after reprocessing with the latest software and using the displacement as a measure, the exposure subsets did not exhibit the previously observed difference in the motion.
10. Increased the thickness of the green contours in Supplementary Figure 7
11. Added data processing workflow trees for all comparative subsets as Supplementary Figures 8 - 12

Response to reviewers

We are grateful to the reviewers for their positive comments and constructive criticism. For the revision, we took into account all suggestions and tried to answer all reviewers' questions. We believe that this greatly improved the quality of the manuscript and made it more transparent. Following is a point-by-point response to the reviewers.

Reviewer #1 (Remarks to the Author):

Here, the authors provide an in-depth investigation into the influence of sample preparation and data collection parameters on the ability to routinely determine 3 Å or better resolution cryo-EM structures of GPCR specimens. The manuscript is well-written and the data are clearly presented. The systematic dissection of the contributing parameters during EM grid preparation and data collection are a worthy contribution to a rapidly-evolving field where such parameters are easily influenced by anecdotal evidence and subjective preferences. This reviewer only has a few minor criticisms, mostly pertaining to the data processing and evaluation of the final particles.

We appreciate the reviewer's positive outlook of our work.

Comments/criticisms:

-Based on the evidence provided in the manuscript, particularly Figure 3, there are still several structures (post Aug 2019) that are not resolved to the 2.5 Å resolution threshold the authors reference. Although some are unpublished, GLP1R (PF-noNb) represents a published data set that was not resolved to 2.5 Å or better. I am curious if all of the optimized parameters outlined in the manuscript were used for this data set? If not, which parameters were not utilized? Perhaps there is more information to be gleaned about why this GLP1R sample did not perform as well as the others.

The GLP-1R-PF-noNb was indeed acquired using the optimized experimental conditions. However, this complex did not include the stabilizing nanobody 35 and therefore was more dynamic. The details were discussed in our recent preprint on structure determination without the Nb35 (Xin Zhang et al. "Evolving cryo-EM structural approaches for GPCR drug discovery", bioRxiv (2021)). To emphasize that there are still challenges related to the type of sample we added the following paragraph to the discussion:

"Other factors, that are outside of the scope of this study, will also have a substantial influence on the performance. The size and flexibility of a protein complex are fundamental attributes for cryo-EM. Smaller and/or more dynamic molecules will generally produce lower resolution reconstructions. Recent examples are the PF 06882961-GLP-1R-DNGs³⁹ that did not include the proprietary stabilizing nanobody 35 (shown in Fig. 3 as GLP-1R (PF-noNb)), and two CGRP receptor-only structures⁴⁰ that were less than half the size of a ternary GPCR-G protein complex. For such challenging targets to become routine and improve in resolution, basic cryo-EM technologies must be advanced further. In this respect, direct detectors continue to evolve, and recent innovations include an electron even representation (EER) format that mitigates the limitation of fixed movie frame width by recording every individual electron event⁴¹. This and other developments will contribute to achieving better signal-to-noise ratio and higher throughput that will provide significant benefits for difficult samples."
"

-Although the authors use a series of metrics to evaluate the final maps, the "alignability" of the particles was not evaluated. Specifically, parameters outputted by RELION (e.g., `rlnNRSignificantSamples`, etc.) that might further inform on the effects of data collection parameters outlined here. I find the lack of such analyses surprising given the rigor in the paper. Perhaps the authors already looked into such metrics but did not find anything conclusive worth reporting? In particular, I am curious how histograms of such values compare for a given protein across data collection parameters and if certain data collections schemes

Following the reviewer's suggestion, we added histograms of the `rlnNrOfSignificantSamples` metric for all subsets in a new Supplementary Figure 3 and calculated the median and standard deviation of the distributions. We also included comparison plots of the angular (`rlnAccuracyRotations`) and translational (`rlnAccuracyTranslationsAngst`) accuracies as panels e and f in Supplementary Figure 2. These metrics were also included in Table 2. Overall, the alignment statistical values recapitulated the results from the B-factor statistics, but there were a few interesting observations about the VPP, the objective aperture and the defocus, and we added them to the text:

"Curiously, the reported angular accuracy for the phase plate data was slightly better (Supplementary Fig. 2e), but the rest of the alignment statistics favored the defocus data (Supplementary Figs. 2f and 3a)."

"The aperture did improve slightly the alignment performance of the particles (Table 2, Supplementary Figs. 2e, f and 3c)"

“Overall, the lower defocus subset did not show significant performance deficits, despite the lower contrast in the images. Nevertheless, alignment accuracies, sharpening B-factor, and possibly particle picking and classification were affected by the lack of low frequency components (Table 2, Supplementary Figs. 2e, f, 4d and f).”

“... we re-polished the particles with a larger 332 Å box ... and the number of significant samples distributions changed in favor of the higher defocus (Supplementary Figs. 3d and f).”

-The number of frames used during movie recording was increased for the GLP1R data sets, compared to PAC1R, which were consistently resolved to higher resolution. Given the impact of early motion correction to the final resolution, do the authors think the shorter frames also benefited frame alignment and recovery of high-resolution information for the 3D reconstruction?

Before March 2019 we used ~ 1 e/Å² per frame in our data acquisition, from March of 2019 we started using ~ 0.85 e/Å² per frame. This may have had a small positive effect, but we do not have comparative data to properly quantify it. Nevertheless, we added a short paragraph about pixel size and dose per frame to the discussion.

“The pixel size and exposure per frame are also important experimental parameters. We do not have comparative data for them, but we usually aim for the pixel size that is closest to one third of the target resolution, e.g. $2.5 \text{ \AA} / 3 = 0.83 \text{ \AA}$. This places the target resolution at two thirds of the physical Nyquist frequency of the detector. With densely packed smaller particles, it may be beneficial to go to a smaller pixel size that is a quarter of the target resolution, placing it at half of the physical Nyquist periodicity. For the exposure per frame, we aim for ~ 0.8 e/Å²/frame. “

-Surprisingly, the authors did not provide any data processing trees. Although these proteins were previously published, presentation of the data processing trees would be beneficial and help inform on the quality differences at each stage of 3D classification. In particular, it would help to know if the same 3D classification parameters (i.e., tau fudge, etc.) values were used for each successive 3D classification or if they were varied? And why? Which values did the authors settle on?

We apologize for not including data processing trees in the original manuscript. We added four supplementary figures (Supplementary Fig.8-12) with processing workflow trees for all subsets and updated the methods section accordingly. For reviewer’s reference, we also attach the original processing trees from the initial submission and preprint version of the manuscript. In all 2D and 3D classifications we use the default tau fudge parameter of 2 for 2D and 4 for 3D. We have experimented with the tau parameter in the past, but did not find higher values to be beneficial for classifications, in particular for GPCRs. We added the following sentence to the methods:

“We used default regularization parameter (tau fudge) values of $T = 2$ for 2D and $T = 4$ for 3D classifications.”

- Can the authors comment on why they did not use 2D classification for pruning the dataset once template-based picking was employed? I find this a bit perplexing and am curious which step in the 3D processing benefits the removal of particles exhibiting template model bias?

In our experience, 3D classification is more effective for cleaning of datasets. It provides more reference projections for particles to find their orientation and is less dependent on the human factor in the selection of good classes. To minimize the propagation of reference bias, we use a 20 Å low-pass filtered reference and “fast subsets”. For post-3D-refinement classification, we used to use a non-alignment 3D classification, but recently found that small local searches (same steps as in the last iteration of Refine3D) improve the performance by allowing lower quality particles to “wobble-out” of

alignment and move to lower resolution classes. We also added the following sentences to the methods:

“All subsets were cleaned solely by 3D classification, which in our opinion is less-prone to operator bias and provides more consistent results.”

“The particle stack was then further cleaned through a 3D classification with 5 classes and small local searches (same search steps as in the last iteration of the previous 3D auto-refine), which in our experience works better than no-alignment 3D classification for sorting-out low-quality particles. “

-I find the lack of FSC curves in the manuscript surprising. Although some of the reported resolution values are statistically insignificant, differences in the FSC curves could be informative. Furthermore, Guinier analyses could provide even further information about the differences in quality between maps when the reported resolution values are minor.

We apologize again, this time for not including FSC plots in the original manuscript. We added Supplementary Figure 4 with FSC plots for all particle subsets. In the legend of each plot we included the estimated resolution and the B-factor that was determined from the Guinier plot fits. We hope that this is the value that the reviewer meant by “Guinier analyses”. Overall, the sharpening B-factors recapitulate the Rosenthal-Henderson B-factor conclusions, with the exception of the defocus magnitude subsets, where the sharpening B-factor was lower for the higher defocus subset. We added the following sentence to the text:

“Nevertheless, alignment accuracies, sharpening B-factor, and possibly particle picking and classification were affected by the lack of low frequency components (Table 2, Supplementary Figs. 2e, f, 4d and f).”

-I wonder if the authors could comment on the magnification selected for imaging and whether this was also optimized similar to the other data collection parameters? Seems to be a perfect fit for the resolution range of most structures but curious if slightly higher mags were also tested.

Magnification and the related pixel size are indeed quite important parameters. We have not done any systematic optimizations in this direction and have used 0.83 Å/pix for most of our datasets. We used a smaller pixel of 0.65 Å in some recent datasets, but with mixed success. For a couple of samples this allowed us to reach sub-2 Å resolution, but for other, more heterogeneous ones, it limited the number of particles and probably also the final resolution. As mentioned above, we included a short comment about the pixel size in the discussion.

-Density subtraction of the detergent micelle is a common practice for small membrane proteins but it doesn't appear the authors employed this. Has it been attempted but not beneficial?

A few years ago, we did attempt density subtraction of the micelle but it did not improve the results. Instead, we find that 3D auto-refinement with a smooth mask that excludes the micelle works quite well. We have also tried the recently released SIDESPLITTER and it produced basically the same results as masking-out the micelle.

Minor:

Fig. S4. The green lines for the polygons are very difficult to see.

We made the lines for the polygons and the grid squares thicker.

Table 1: “phase palte” should be corrected

We corrected the typo.

-Mark Herzik

Reviewer #2 (Remarks to the Author):

It has been a real pleasure to read the manuscript by Danev et al., as I have been closely following this collaboration for years. It has been truly pioneering work in the field of cryoEM structure determination of GPCR-G protein complexes and they have had many milestone achievements.

While the technique of cryoEM has made remarkable contributions to biological research as a whole, ideal experimental parameters are typically very hard to find, are usually just propagated word-to-mouth, and are usually of anecdotal nature. The manuscript shows hard, experimental evidence for best practices in cryoEM, which can not only be applied to GPCRs. Besides, Prof. Danev has spent countless years on the development of the VPP and I applaud him for his honest evaluation of the contribution of the VPP on data quality.

The manuscript has been uploaded to bioRxiv in August and has been very well received by the community. The paper has been downloaded over 2000 times and it already has been cited twice, which is remarkable. The manuscript by Danev et al. will be an invaluable reference for training the next generation of electron microscopists. It can be considered an “instant classic” and I highly recommend publication with minor revisions.

We thank the reviewer for their positive comments about our work.

The only substantial questions and remarks I have are regarding the data processing schemes. I have spent quite a bit of time looking at the supplementary text and table 2:

- From the supplementary text alone it is unclear, at what point exactly the datasets were split. A flowchart, highlighting the key differences, and at what point the datasets were split, would be helpful to keep track of things. I have attempted to assemble such a flowchart (see below). From my understanding picking was done on the ‘full’ datasets (except the ‘low dose’ dataset?). The +VPP/-VPP, low dose/high dose and +ZLF/-ZLF were processed separately after picking (3D class, ctf refinement, etc.), while +OLA/-OLA and high/low defocus were separated only at a later stage (after 3D classification, CTF refinement, polishing). This might well be a misunderstanding on my end, but I cannot really follow the reasoning behind this discrepancy and would be interested to hear the authors justification for not treating every dataset ‘equally’, meaning either treat all datasets completely separately, or split them at the same point. I could well imagine that some of the results shown (or the lack of a significant effect) could be influenced by this difference in treatment.

We apologize for not including data processing trees in the initial submission. This was a serious omission on our part. We would also like to thank the reviewer for the criticism, which encouraged us

to reprocess all data using completely separated subsets from the very beginning. For the reprocessing, we used an identical workflow for all subsets and equalized the number of micrographs in the comparative subsets. It was a simplified version of the optimized workflows that were used for the original processing. We added Supplementary Figures 8-12 with the new processing trees and attach for reference the original processing trees.

Originally, the VPP/defocus, ZLF/no ZLF and the 65/40 e/Å² particle subsets were processed independently, but with slightly different “optimized” workflows. The OLA and DEF subsets were indeed separated from the full GLP-1R-GLP-1 dataset by extraction from the original micrographs, which we believed would erase any bias from the global processing of the complete GLP-1R-GLP-1 dataset. The results from the new independent processing of the subsets showed that the reviewer’s concern about the post-global reconstruction splitting of the data was justified. Originally, the differences between the +OLA/-OLA and the +DEF/-DEF subsets were negligible. After independent processing, the subsets show more clear differences, with still very small but noticeable positive effect of the OLA, and a much clearer improvement for the higher defocus data with a larger box. The effects of the VPP and higher total exposure were also enhanced by using the same processing workflow without special tuning for individual subsets. We updated the main text and the discussion to reflect the newly calculated results.

- Picking: This mostly is a continuation of my previous point. It appears that the number of picked particles per micrograph varies between the ‘half’ datasets (by 10-25%), and as stated above in some cases picking was done on the consensus dataset, in some cases individually on the split datasets. The difference appears to be smaller for the ‘consensus treatment’ (i.e. OLA and defocus), while separately treating datasets resulted in larger differences in picked particles per micrograph (VPP, ZLF and dose). This difference could very well arise from superior data quality with the beneficial experimental parameter, but it is an important caveat to mention that one might be able to compensate (or at least influence the result) by tweaking the parameters towards the ‘weaker’ data. It could theoretically also point to sample variation, which would introduce artifacts in the interpretation. Another complication regarding picking is that the centering for each dataset could be very different. It would be helpful to report average ‘_rlnOriginXAngst/_rlnOriginYAngst’ after 2D classification, which reports the distance from the original center to the refined center, to show that the “quality” of the centering was comparable.

We agree that picking could be a point of discrepancy. In our workflows we usually overpick by a good margin, as judged visually by having some picks in empty ice areas. We also always use an internally derived 3D reference for the final picking. With a reference, the particle offsets are typically just a few pixels. Furthermore, re-extraction (if “re-center” selected) and Bayesian polishing re-center the particles according on their refined coordinates. The main challenge in this work was to match the number of particles in the comparative subsets. We did our best to try to match the particle numbers by adjusting the picking threshold in Relion. Nevertheless, there was a general tendency of more picks in lower contrast micrographs and with a higher quality internal reference (e.g. EXP subsets). We did not include comments about the picking performance because it is very much software package, version and algorithm dependent, and is beyond the scope of this work.

- Defocus: The notion that low defocus values used in data collection result in better reconstructions, has gained traction in the recent past. However, a recent publication by Glaeser et al. (Ultramicroscopy) has questioned this “dogma” and suggests defocus values as high as 5 um to be ideal. It would be worth commenting on this in the paragraph about the effect of defocus values on reconstructions. To underline the positive effect of smaller defocus values, it would be possible to report the results of more than one dataset, since all datasets have the same total range of -0.7 to -

1.7 μm . There is also mention of a 'PAC1R defocus subset' in the supplementary text, but no further discussion about it, as far as I can tell.

We added a reference to the recent paper by Glaeser and colleagues and agree that the low defocus dogma is a misconception. Indeed, the results from the independent processing of the defocus subsets showed that the higher defocus subset performs slightly better with a larger box. However, with a typical box size of 1.5 times the particle diameter, the low defocus subset produced a slightly higher resolution total reconstruction. Therefore, we still consider it beneficial to include lower ($<1 \mu\text{m}$) defocus values in the defocus range from a data processing throughput point of view. The requirement for a larger box at higher defocus can become a practical bottleneck with the current software packages, especially when the resolution is close to or below 2 \AA .

The "PAC1R defocus" subset refers to the non-VPP subset of the PAC1R dataset. We may indeed revisit the defocus performance question in the future with datasets that reach below 2 \AA and hopefully with new more-optimized data processing strategies. For this work, we believe that the conclusions are well-supported by the presented results and additional data processing with the current approaches will not give different outcomes.

- Masks: Masking is known to have a significant influence on resolution estimates. While I am sure that the authors have paid great attention to using masks with similar 'features', it would be worth mentioning how they were generated (B-factors, smooth edges, etc.), to ensure that the two compared datasets were treated equally.

In our experience, softer masks produce better results during 3D refinement. We typically use 5 pixel expansion with 10 pixel soft edge masks. In the original processing we used tighter masks in the post-processing steps. For the reprocessing here, we used the same mask as in the Refine3D for the post-processing. We were careful to make similar masks between the comparative subsets by using the same threshold and checking the maps for the presence of residual micelle density when selecting the higher threshold for the no-micelle masks. We added the following sentences to the Methods:

"All post-processing steps used the same mask as the corresponding 3D auto-refinement step. Masks were prepared based on the 3D map from the preceding 3D auto-refinement or 3D classification step using a threshold that either included or excluded the micelle. The mask parameters and the inclusion of the micelle are indicated in the processing workflows (Supplementary Figs. 9-12)."

Other minor comments:

- "Beam-induced motion statistics showed a clear reduction in the early motion with 37 the Au foil grid (Fig. S5A) but similar total motion (Fig. S5B)." This is a very interesting and unexpected finding. This implies that the positive effect of using gold grids is only with respect to movement on the first frames, but not the overall doming of the ice. It would be very interesting to plot a comparison of the per frame motion, as I would guess the motion on the gold grids would be more uniform (or something more like a gaussian?).

We thank the reviewer for bringing attention to the motion statistics. Indeed, the original results raised interesting questions. Upon closer examination, we realized that the motion statistics reported by Relion are dominated by noise because they represent the sum of absolute inter-frame shifts and with higher number of frames these are quite noisy. Furthermore, the results in Fig. S5C were obviously unrealistic because the 40 e/\AA^2 subset should present total motions that are smaller than the longer exposure complete set. In the revised manuscript we used the displacement instead of the accumulated motion (Supplementary Fig. 6). It showed a clear difference in behavior between the carbon and the

gold foil grids. In addition, the 40 e/A² subset initial motion statistics were identical to the initial motion of the 60 e/A² exposure. Therefore, we removed panel c from the supplementary figure and the associated comments from the text. We added the following description to the Methods:

“In the preprint version of the manuscript²⁶ we used the early and total motion values reported by Relion, which represent the total motion distance calculated as the sum of the absolute shifts between frames. However, when the beam-induced movements are small, and the number of frames is large, the motion distance values tend to be dominated by the noise in the inter-frame shifts. Therefore, here we chose to use the displacement between the first frame and the frame corresponding to ~4 e/Å² exposure (frame No. 5 for the PAC1R and frame No. 6 for the GLP-1R-TAS and GLP-1R-GLP-1) for the early displacement and between the first and the last frame for the total displacement. These statistics may downplay non-monotonous (e.g. U-turn) beam-induced movements, but showed a very clear motion magnitude difference between the carbon and gold foil support grids (Supplementary Fig. 6).”

- ZLF: From what I understand, the energy filter alone without using the slit (i.e. ‘-ZLF’) can introduce artifacts and could potentially produce data of inferior quality compared to a system without any energy filter. This caveat would be worth mentioning in either main text or the supplementary.

We assume that the idea that a filter without slit would perform worse than no filter is most probably related to non-achromaticity, where inelastic signals will be displaced relative to the primary elastic image. We believe that this could be of significance in materials science, where the sample consists of well-oriented atomic columns which can produce features in the inelastic image. With amorphous biological specimens observed at low doses, we do not expect any contrast in the inelastic components and therefore they can be assumed to be just Poisson noise. Non-achromatic shifts of noise will have no additional detrimental effects, apart the noise itself which is also present in the non-filter configuration. In this regard, our recent performance testing of a newly installed microscope equipped with two cameras, one before and one after an energy filter, showed no performance difference between the before the filter and the filter without slit results:

<https://academic.oup.com/jmicro/advance-article/doi/10.1093/jmicro/dfab016/6273016>

- “The recently introduced M package overcomes this limitation by performing a CTF sign correction on a larger area before extracting the particle box.” It is fair to state, that most ‘classic’ data processing packages (IMAGIC etc.) used to correct for the CTF before extraction.

We added a reference to IMAGIC to the sentence:

“Similar to processing approaches from the early days of cryo-EM³⁹, the recently introduced M package overcomes this limitation by performing a CTF sign correction on a larger area before extracting the particle box³².”

So long.

Cornelius Gati

Reviewer #3 (Remarks to the Author):

This manuscript describes a comparative study of using different specimen grid preparation and electron microscope settings to collect images of three different single particle specimens of G protein receptor membrane proteins. The resolution differences between maps resulting from different settings are within a fraction of Å. The results are not surprising to experienced electron microscopists though they may be useful to novice users. One of the shortcomings of the manuscript is nothing conceptually new. But the studies were done well in each comparison. I noted that some relevant literature was missing in the current manuscript. It would have been better if the comparison was done with a single specimen with various combinations of parameter choices. The followings are my comments on each of the experimental variables used in the study.

We thank the reviewer for their comments. We agree that the paper would probably be of utmost interest to the large number of new cryo-EM users whose main interest is getting the highest quality structure of their sample. In fact, this was one of the main motivations for this study. Notwithstanding, we are not aware of other systematic studies of the effect of experimental parameters on the cryo-EM performance for membrane proteins.

Volta phase plate (VPP): it is known that Volta phase plate yields impressive low resolution image contrast but does not yield higher resolution. This is why not too many reported structures using VPP except the first author who invented the VPP. It has been shown in the literature that there is no practical advantage to use the VPP from the point of view of pursuing high resolution studies for small biomolecules (e.g., G. Lander). However, PP is still of meritorious value in a recent study (e.g., R. Glaeser).

Indeed, the advantage of using the VPP has been questioned since its inception. The GPCR results it produced in the past were of great value, which gave many researchers the impression that it is beneficial. Here, we show quantitatively that in state-of-the-art cryo-EM it does not provide any benefits.

Energy filter: there was an early study to discuss the physical rationale of energy filter in affecting the image contrast (e.g., J Langmore). It is generally agreed that energy filter is useful for thick specimen or thick embedding ice. There was a recent atomic resolution structure report which was done with and without energy filter in two different electron microscopes and cameras (e.g., W. Chiu). I agree that the current study is well-executed to show the difference with and without energy filter. However, the difference in resolution is not substantial in general.

We thank the reviewer for their positive comments. Following reviewer #2's suggestion we reprocessed the datasets through an identical workflow for all subsets and this increased the performance advantage of zero-loss energy filtering to ~ 0.25 Å.

Objective aperture: It has been known that objective aperture is critical for obtaining higher quality low dose images (e.g., PNT Unwin). However, it is nice to see that there is not much difference in the current generation of instrumentation.

After reprocessing of the subsets, we observe a slight performance advantage from using the objective aperture, but we warn that it should be used with caution because of potential unpredictable aberrations due to electrostatic charging of the aperture, especially at resolutions below 2 Å.

Defocus setting: It is well known from the phase contrast theory that closer to optimum defocus setting is the preferred choice if high resolution is pursued. What is shown agrees with expectation.

The optimal defocus question has remained open in cryo-EM and continues to evolve. Here, we tried to determine if the low defocus setting has disadvantages for GPCRs. We conclude that its performance is very close to that of higher defocus while being easier to process.

Electron dose limits retrievable high resolution structure features due to radiation damage. This result is interesting. In rationalizing it, the observed phenomenon may be attributable to the proper weighting of the movie frames. The higher dose data yields better statistically defined particle images which would improve the particle orientation estimation accuracy and thus produces a better resolution map.

We agree that correct dose weighting is essential and that the most probable reason for the benefit from a higher dose is the boost in low frequency contrast.

The following are a few detailed suggestions on the current manuscript

1. In this study, for objective lens aperture testing, the authors chose the 100 micron. Did the authors try the 70 micron aperture? Is it because 70 micron one limits the high resolution on the scope? (page3, line28)

We did not test a 70 μm objective aperture because on a Krios its resolution cutoff is 2 Å and we have been hoping to, and actually recently achieved GPCR resolutions below 2 Å. The 100 μm aperture has a resolution cutoff of 1.4 Å, but we still don't feel comfortable using it because of the very small benefit while having a potential risk for introduction of uncorrectable higher order aberrations due to electrostatic charging.

2. Page4, line 2. This sentence needs to be revised and clarified.

We revised the explanation of the B-factor line shifts to:

“While the slope of the B-factor plot represents the information decay as a function of spatial frequency, the offset of the curve corresponds to the overall signal-to-noise ratio in the data. To compare this quantity, we calculated two additional performance measures from the linear fits - the resolution from 100k particles and the number of particles necessary to reach 3 Å resolution (Table 2, Fig. 2a, c, Supplementary Fig. 2).”

3. In Table 1, “Grid type” showed twice in the table; I am also curious why the glow discharge time is so different (30s vs 90s) on the same type of the grids (Quantifoil R1.2/1.3 Cu/Rh 200). Did the author try other types of the grids rather than R1.2/1.3, such as R2/1 or R3.5/1.

We corrected the repetition. We settled on a longer glow discharge time because it reduced the pooling of sample solution in the middle of grids squares. We have not tried other grid types because in our experience smaller grid holes provide more stable ice. Here is the explanation about the glow discharge time from the grid optimization part of the Results:

“We had to significantly extend the glow discharge time to 90 s (Table 1), to improve grid wettability and enhance sample drainage during blotting. This reduced the pooling of solution in the middle of grid squares, especially with 200 mesh grids (compare Supplementary Figs. 7a and b with 7c and d).”

4. Page4, line 30. Why did the author choose 25 eV energy slit instead of smaller one, since they used 5 eV before and achieved a better resolution structure.

We added a sentence to the Discussion explaining the reason for using a 25 eV slit:

“Here, we used an energy selection slit width of 25 eV due to long-term stability issues with our hardware setup.”

We had problems with energy shift drift which caused the slit to sometime obscure the image during overnight data collection. Recently, we use a SerialEM script which automatically re-centers the slit every ~3 hours

We have not used 5 eV slit for data collection yet. Such results were published by the MRC group in their “Single particle cryo-EM at atomic resolution” paper. We have been using mostly 15 eV slit in our recent experiments.

5. Page6, line 13. The authors mentioned “blot force” was one of the parameters they optimized. This value should be added in Table 1.

We omitted on purpose the “blot force” numbers because in our experience they are relative and the actual offset varies quite a bit between different Vitrobot machines. Our concern is that readers will take the provided values a recommendation.

6. In Fig.3, it is better to mark the same sample using the same shape for better understanding.

The figure is already quite cluttered, and the size of the symbols represent the cryo-EM efficiency value. Therefore, we prefer to keep all symbols as circles and rely on the labels for identification of the samples.

7. Page9, Line23-26, can the author describe a little more details on how to perform the cryoEF and measure the ice thickness instead of just a citation?

We added the following description about the cryo-EF and ice thickness calculations to the Methods section:

“The cryo-EM efficiency (cryoEF) value was calculated on the particle set from the final 3D refinement of each dataset using the cryoEF program ³⁸ with 256 pixels box size with the commands: “extractAngles.sh run_data.star > angles.dat”, where the “run_data.star” file is the particle star file from the final 3D auto-refinement; and “cryoEF -f angles.dat -b 256 -B *Bfac* -D 132 -r *Res*”, where the *Bfac* and *Res* parameters are the sharpening B-factor and resolution from the final post-processing job in Relion. The ice thickness in Supplementary Figure 5 was calculated using the formula $t = \text{MFP} \cdot \ln(I_0/I)$, where the MFP is the inelastic mean-free path, I_0 is the open beam intensity measured from images that were taken in holes without ice, and I is the average intensity of a micrograph. The inelastic mean-free values path used in the formula were 395 nm without an objective aperture and 322 nm with an aperture ⁴⁶.“

REVIEWER COMMENTS

Reviewer #1 (Remarks to the Author):

The authors have extensively addressed the original concerns raised by this author. The additional data points and subsequent analyses were thoroughly performed and seamlessly incorporated into the manuscript.

I have no other major concerns.

Reviewer #2 (Remarks to the Author):

I truly appreciate the heroic effort of the authors to reprocess all datasets, and the general improvements of their supplementary materials. I see no need for further changes and recommend the publication as is.

Cornelius Gati